# Application of Plasma Activation in Flame-Retardant Treatment for Cotton Fabric

**DOI:** 10.3390/polym12071575

**Published:** 2020-07-16

**Authors:** Huong Nguyen Thi, Khanh Vu Thi Hong, Thanh Ngo Ha, Duy-Nam Phan

**Affiliations:** 1School of Textile—Leather and Fashion (STLF)—Hanoi University of Science and Technology (HUST), No. 1, Dai Co Viet, Hai Ba Trung, Hanoi 100803, Vietnam; huongnt87@hict.edu.vn (H.N.T.); thanh.ngoha@hust.edu.vn (T.N.H.); nam.phanduy@hust.edu.vn (D.-N.P.); 2Hanoi Industrial Textile Garment University, Le Chi, Gia Lam, Hanoi 100803, Vietnam

**Keywords:** atmospheric-pressure DBD plasma, flame-retardant, cotton fabric, Pad-dry-cure method, LOI, tensile strength, Pyrovatex CP New, Knittex FFRC

## Abstract

Cotton fabric treated by Pyrovatex CP New (PCN) and Knittex FFRC (K-FFRC) using the Pad-dry-cure method showed an excellent fire-retardant effect. However, it needed to be cured at high temperatures for a long time leading to a high loss of mechanical strength. In this study, atmospheric-pressure dielectric barrier discharge (APDBD) plasma was applied to the cotton fabric, which then was treated by flame retardants (FRs) using the pad–dry-cure method. The purpose was to have a flame-retardant cotton fabric (limiting oxygen index (LOI) ≥ 25) and a mechanical loss of the treated fabric due to the curing step as low as possible. To achieve this goal, 10 experiments were performed. The vertical flammability characteristics, LOI value and tensile strength of the treated fabrics were measured. A response model between the LOI values of the treated fabric and two studied variables (temperature and time of the curing step) was found. It was predicted that the optimal temperature and time-to-cure to achieve LOI of 25 was at 160 °C for 90 s, while the flame-retardant treatment process without plasma pretreatment, was at 180 °C and 114 s. Although the curing temperature and the time have decreased significantly, the loss of mechanical strength of the treated fabric is still high. The tensile strength and scanning electron microscopy (SEM) images of the fabric after plasma activation show that the plasma treatment itself also damages the mechanical strength of the fabric. X-ray photoelectron spectroscopy (XPS) spectra of the fabric after plasma activation and energy-dispersive spectroscopy (EDS) analysis of the flame retardant-treated (FRT) fabric clarified the role of plasma activation in this study.

## 1. Introduction

Cotton is one of the most abundantly used fibers. Nevertheless, it is one of the most flammable fibers as well with low LOI of 18.4% and the onset of pyrolysis at 350 °C [1]. Hence, the application of FR products on cotton is an important textile issue. Many studies have attempted to impart flame retardancy in cotton by grafting FR groups or coating with FR layers [2]. Recently, X. Hu [3,4,5] has used inorganic compounds in water-based fireproof nano coating for the steel and plywood. The results showed that the flame-retardant efficiency of these coatings was better, however, they have not been tested for the cotton fabric, especially for the purpose of making the durable flame-retardant cotton fabric.

Among the durable FRs, one of the most commercially successful known under the trade name of Pyrovatex CP New is a N-methylol dimethyl phosphonopropion amide (MDPA). Its reaction with cellulose hydroxyl is shown in the following scheme: R-NHCH_2_OH + HO–Cell -> R-NHCH_2_O-Cell+ H_2_O [6]. Moreover, the laundering durability of cotton fabrics treated by PCN can be improved with the resin applications. Organophosphorus agents are combined with melamine resins and an acid as catalyst, i.e., phosphorus acid, but phosphoric acid may decrease the tensile or tear strength of cellulose fabrics [6,7,8,9,10]. A further negative consequence of the usual organophosphorus compound is formaldehyde release, in particular, when trimethylolmelamine (TMM) resin is used [6,9]. To minimize these disadvantages, replacing TMM with other crosslinking agent has been the solution of many researchers. In those studies [11,12], 1,3-dimethylol-4,5-dihydroxyethylene urea (DMDHEU) has been used as a crosslinking agent with low formaldehyde content [11,12]. However, with DMDHEU, the curing steps were also required at high temperatures (170 °C), and it also demands phosphoric acid as catalyst. Polycarboxylic acids (1,2,3,4- butane tetracarboxylic acid and citric acid) have also been considered as more appropriate options because with their carboxylic groups, they can produce a cross linkage between cellulose and PCN. Moreover, they are the formaldehyde-free crosslinking [2,6,7,9,13]. However, the mechanical strength loss of the treated fabric was also important, which was due to the high temperatures of curing (180 °C) and the low pH of the padding solution. In our previous study [2,14,15], the modified dihydroxy ethylene urea (DHEU) labeled (K-FFRC) had been used as a formaldehyde-free cross-linking agent between celluloses and PCN. The results showed that the treated fabric is flame retardant (LOI > 25). Moreover, the laundering durability of the treated fabric was also good (LOI of the treated fabric after 30 wash cycles was always more than 25 [2,14,15]). Moreover, the free formaldehyde content of the treated fabric was acceptable. Therefore, DHEU crosslinking seems to be a better solution than the ones mentioned above. However, with K-FFRC, the cotton fabric still had to be cured at 180 °C, so the mechanical loss of the treated fabric was still high (26.9% [2] and had been varied from 18% to 32% [14]).

How to reduce the curing temperature and still get the durable flame-retardant cotton fabric? The surface treatment for the fabrics before the fire-retardant treatment is a solution?

Surface modification of textiles is of great importance, because it allows to enhance some necessary properties of textiles such as softness, dyeability, adhesion and wettability. They can be chemical modification or physical modification [16]. Of the different kinds of surface modification techniques applied, plasma treatment appears to be the most commonly used with good results and environmental benefits [16,17,18]. Plasma treatment is a physicochemical method used for surface modification because it affects the surface both physically and chemically without altering the material bulk properties [19]. The principle of plasma surface modification is as follows: The plasma atmosphere consists of free electrons, radicals, ions, atoms, molecules and different excited particles depending on the plasma gas used. It is the interaction of these excited species with solid surfaces placed in the plasma reactors, which results in the chemical and physical modification of the material surface. All of the active species react with the substrate surface, this creates chemical functionality on the substrate surface [18]. Moreover, the formed reactive particles react in a direct way with the surface without damaging the bulk properties of the treated substrates. In fact, the surface modification is limited to the outermost 10 to 1000 Å of the substrate [18]. The plasmas can be classified as being of the low pressure and atmospheric type. Both plasmas can be used for the surface activation, however, atmospheric plasma has many advantages when compared with vacuum plasma. Atmospheric pressure plasma, which is generated in ambient air and achieves continuous in-line material processing at high speed [19,20,21,22]. Among four main types of atmospheric plasmas applied to textiles, the dielectric barrier discharge (DBD) technology in the air is one of the most effective nonthermal atmospheric plasma sources and has been attracting increasing interest for industrial applications due to its scalability to very large systems [22]. DBD plasma is able to generate a high flux of the reactive and metastable species, it can be used for the large-scale treatments. It is a class of plasma source that has an insulating (dielectric) covers over one or both of the electrodes and operates with high voltage power [22,23,24]. This presence characterizes DBD over other types of plasma, the plasma in DBD is cold because of the presence of the dielectric layer, which limits the heating current in the circuit while the displacement current has not any heating effects [24]. Thanks to these advantages, DBD plasma has been used for the surface activation before different functional finishes [19,23,24,25,26,27,28,29,30].

For these reasons, in this study, the APDBD plasma was applied on the cotton fabric before it was treated with FRs using the pad–dry-cure method. It is hoped that the plasma treatment could make the cotton fabric more active, so the reaction between cellulose and chemicals (PCN and K-FFRC) could occur more easily. As a result, the curing temperature and time can be reduced, but it must be ensured that the treated cotton fabrics will be durable flame-retardant fabrics. The purpose is to have a flame-retardant cotton fabric (LOI ≥ 25) [1] and the mechanical loss of the treated fabric due to the curing step is as low as possible, this means that the curing temperature need to be as low as possible and the curing time as short as possible. The effect of the plasma activation on the cotton fabric will be evident by comparing the vertical flammability characteristics, LOI value and tensile strength of the FRT samples with plasma preactivation and without plasma preactivation. The new point of this study is to clarify the impact of the APDBD plasma pretreatment on the effectiveness of the flame-retardant treatment for cotton fabrics by PCN in combination with K-FFRC. The observed phenomena were all explained by analyzing SEM images, XPS spectra and EDS spectra of samples before, after the plasma treatment and after the flame-retardant treatment.

## 2. Materials and Methods

### 2.1. Materials

Fabric: The 100% cotton fabric, desized, scoured, bleached and mercerized was supplied by NATEXCO (Nam Dinh, Vietnam). It is 3/1 twill weave fabric with construction of 28.3 × 35.8 (Tex)/133 × 61 (per inch) weighing 242 g/m^2^.

Chemicals: PCN, K-FFRC, invadine PBN were supplied by Huntsman (provided by Hong Phat New Trading company limited, Hanoi, Vietnam). PCN is an N-methylol dimethylphosphine propionamide, in this study, it was used as a flame-retardant agent. K-FFRC is a modified dihydroxy ethylene urea, it was used as a cross-linking agent. invadine PBN was used as a tenside surfactant. (The scheme showing the crosslinking mechanism of DHEU between PCN and cellulose is presented in [2]).

### 2.2. Method

Cotton fabric is treated according to the procedure described in Figure 1.

#### 2.2.1. APDBD Plasma Treatment for Fabric

Laboratory APDBD plasma equipment with a width of 50 cm developed by School of Engineering Physics (SEP) of HUST as part of the KC.02.13/16–20 project was used in this study. The DBD plasma was produced between two parallel electrodes. Both electrodes are covered by a polycarbonate sheet as a dielectric layer. The thicknesses of the upper and lower sheets are 5 and 3 mm, respectively. The length and width of the electrodes are 8 cm and 50 cm, respectively. The cotton fabric with a width of less than 50 cm can move continuously between the top and bottom electrodes. The tension rollers are placed before and after the electrodes to keep the fabric in uniform tension. The movement speed of the fabric can be controlled by a motor. In this study, the width of the cotton fabric was of 35 cm, the distance between the polycarbonate sheets (discharge gap) was 3 mm, the plasma treatment power was 400 W (1 W/cm^2^**)**, an air atmosphere was employed, the plasma exposure time was 90 s (the fabric moving speed of 5 cm per 1 min), unchanged for the all experiments.

#### 2.2.2. Flame-Retardant Treatment for the APDBD Plasma-Treated Cotton Fabric

A bath pad–dry-cure method was used for the flame-retardant finishing. A set of experiments was carried out to find the lowest curing temperature and the shortest curing time while ensuring that the treated fabric will be the flame-retardant. In these experiments, the condition of all steps was unchanged except for the condition of the curing step.

Based on the results of our previous studies [14], in this study, the curing temperature (X_1_) was studied in the range of 160 to 180 °C and the curing time (X_2_) from 60 to 120 s. The central composite designs type face centered (CCF) was used to design the experiments. According to the CCF, the factors were tested at 3 levels (minimum, middle and maximum) of the absolute values (X_1_, X_2_) equivalent to levels − 1, 0 and +1 of the coded values (A, B). The total number of the trials N based on the number of the design factors k, N = 2^k^ + 2k + n_c_ [31] (2^k^ factorial trials, 2k axial trials and n_c_ center point trials (n_c_ = k)) so N = 10. The details of these ten experiments are present in Table 1.

Finishing solutions included 450-g/L PCN, 108-g/L K-FFRC, 5-g/L invadine PBN (this recipe was used in our previous study [14]).

The APDBD plasma-treated fabric samples 35 cm × 35 cm were impregnated with the finishing solution, then padded with the wet pick-up of approximately 80% by padder SDL D394A. The padded samples were dried at 110 °C for 5 min. After, these samples were cured at varied conditions according to the options presented in Table 1. Stenter SDL D398 was used for the drying and curing steps. Next, the samples were rinsed under the running water for 5 min to remove all the residual FRs on the fabric surface (unable to react with celluloses) and neutralize the treated fabric. Then, the fabric was dried in the stenter at 110 °C for 3 min. After that, the treated samples were stored in the polyethylene bags at the standard laboratory conditions for 24 h before any further analysis.

#### 2.2.3. Characterization of the Control and Treated Cotton Fabric

● Flammability test

The vertical flammability test method ASTM D 6413– 2015 was used for evaluating the flammability of the untreated and finished samples.

The LOI value of the control and finished samples were measured in accordance with the ASTM D 2863–97 standard method.

● Tensile properties of fabrics

The ISO 13934–1:2013 standard method was used to determine the maximum force (F_max_) of the control, APDBD plasma-treated and FRT samples. The tensile-testing machine Tenso Lab 2512A, Mesdan Lab (STLF, Hanoi, Vietnam) was used for these tests. The experiments were conducted on the samples with dimensions of 200 ± 1 mm x 50 ± 0.5 mm. The mechanical strength loss of the fabric due to the APDBD plasma treatment and flame-retardant treatment was calculated according to Equation (1).
(1)$$\;\mathrm{Mechanical}\;\mathrm{strength}\;\mathrm{loss}\;{\left(\%\right)}^{}=\frac{{\;F}_{max}\;\mathrm{of}\;\mathrm{control}\;\mathrm{sample}\left(N\right)-{\;\mathrm{Calculated}\;F}_{max}\;\mathrm{of}\;\mathrm{treated}\;\mathrm{sample}\;\left(N\right)}{{\;F}_{max}\;\mathrm{of}\;\mathrm{control}\;\mathrm{sample}\;\left(N\right)}100$$

The density of the FRT samples were changed due to the shrinkage during the flame-retardant treatment. Therefore, the calculated F_max_ of the FRT samples in Equation (1) is calculated based on the number of yarns of the control sample and the number of yarns of treated sample, Equation (2).
(2)$${\mathrm{Calculated}\;F}_{max}\;\mathrm{of}\;\mathrm{treated}\;\mathrm{sample}\;{\left(N\right)}^{}={\;\mathrm{Tested}\;F}_{max}\;\mathrm{of}\;\mathrm{treated}\;\mathrm{sample}\;\left(N\right)\frac{\mathrm{Number}\;\mathrm{of}\;\mathrm{yarns}\;\mathrm{of}\;\mathrm{control}\;\mathrm{sample}}{\mathrm{Number}\;\mathrm{of}\;\mathrm{yarns}\;\mathrm{of}\;\mathrm{treated}\;\mathrm{sample}}$$

● Surface analyses

SEM and EDS were carried out for the surface morphology observation, elemental analysis, P and N mapping of the cotton fabrics before, after the plasma exposure and after durable flame-retardant treatment. The SEM images show the modification of surface morphology due to the plasma treatment and flame-retardant treatment. EDS was used to demonstrate the presence of FRs on the FRT fabrics. SEM HITACHI TM 4000 Plus (SEP, Hanoi, Vietnam) was used for these tests. The EDS (SEP, Hanoi, Vietnam) of the samples was conducted at a beam energy of 15 KeV. The SEM observation was carried out at condition: U = 15 kV, X (magnification) = 4000. The samples were precoated with gold.

Functional groups due to the plasma treatment of the fabric were evaluated by XPS system. The Kratos Axis-Ultra DLD, Kratos Analytical (Shinshu University, Matsumoto, Japan) was used to verify the changes in the surface’s chemical composition of the samples before and after plasma treatment. The pressure in the XPS chamber was reduced to 6 × 10^9^ Torr before samples were excited with monochromatic Al K_α_ 1,2 radiation at 1.4866 KeV.

#### 2.2.4. Statistical Processing of the Experimental Results

The equations between the variables and the experimental results were adjusted in using the Design Expert V 10.0.8 software (STLF, Hanoi, Vietnam).

## 3. Results

### 3.1. The Experimental Results

The experimental results of LOI test, vertical flammability test, tensile test according to the CCF design are presented in Table 2.

### 3.2. Model Determination and Analysis of Fitting Model

From the experimental results, the equation of the response model between the LOI and two variables was output by the Design Expert V 10.0.8. Y1–the equation in actual variables and Z1–in coded variables. The equations and their statistical parameters are presented in Table 3. However, it did not find the response equation between the vertical flammability characteristics and warp tensile strength with the variables.

In Table 3, the coefficient of determination (R^2^) of the equation Y1 is 0.6978. According to the principle of the measure of the degree of fit of the models (Sohail. Y. et al. [10]) a model with R^2^ value above 0.6 is viewed as legitimate or a valid model, thus, the equation Y1 is accepted. Furthermore, the *p*-value is only 0.0152 (less than 0.05), indicating that this model is highly significant according to the principle of significance of the model [31].

The statistical significance of the terms of the model defined by Equation Y1 can be evaluated using the analysis of variance (ANOVA). As well as for the model, a *p*-value of the terms lower than 0.05 indicates that the term is statistically significant, whereas a *p*-value higher than 0.1000 indicates that the term is not significant [31]. Table 4 shows that the *p*-value of the term X_1_ is less than 0.05, which means the coefficient of the term X_1_ is significant. Meanwhile, the *p*-value of X_2_ is 0.0513, higher than 0.05, but it is less than 0.1, so, according to the above-mentioned principle, it can be accepted. As such, all terms of this model are accepted. This model is significant for further analyses.

### 3.3. Effect of the Curing Conditions on the Properties of the Finished Fabric

#### 3.3.1. Effect of the Curing Conditions on the LOI Values and the Optimal Solutions

From Equation Y1, the surface reaction curve related to the LOI values of the FRT samples according to the temperature and time of the curing step is shown in Figure 2.

The response equation (Y1, Z1) and Figure 2 shows that LOI of the FRT fabrics has a linear relationship that is directly proportional to both the temperature and time values of the curing step. Thus, the highest LOI value is related to the sample treated with the highest curing condition (at 180 °C for the 120 s). Under this condition, according to the equation Z1, the predicted LOI of the treated sample can be 29.0 while the experimental value is 28.0 (Table 2). However, if the purpose is only to have a flame-retardant cotton fabric (LOI ≥ 25) with the lowest possible curing temperature and shortest possible curing time, from equation Z1, there may be several options in Table 5.

The empirical results in Table 2 also show that all the FRT samples meet the criteria to be flame retardant fabrics (with LOI greater than 25) except the sample treated at 160 °C for 60 s (A and B equals −1). In our previous study [14] (the flame-retardant treatment process for cotton fabric without plasma pretreatment), in order to get the fabrics with LOI ≥ 25, the fabric had to be cured at least at 170 °C for 120 s or at 180 °C for 114 s. Thus, thanks to the preactivation by the APDBD plasma, in this study, the curing temperature can be reduced from 180 to 160 °C and the time from 114 to 90 s in comparison with our last work [14].

#### 3.3.2. Effect of the Curing Conditions on the Vertical Flammability Characteristics of the FRT Samples

The results of the vertical flammability test of the samples are presented in Table 2 (after-flame time, afterglow time and char length). Figure 3 shows the after-flame time of the samples treated under different curing conditions (FRT samples). Figure 4 shows the images of the FRT samples after the vertical flammability test.

The results show a good effect of flame-retardant treatment on the APDBD plasma-treated cotton fabric. There is a clear difference in combustion behavior between the control, APDBD plasma-treated, and FRT samples in the vertical flammability test. The control and APDBD plasma-treated samples burned vigorously in direct exposure to the ignition source. After removing the combustion source, these samples continued to burn until it had burned out and no char at all (Figure 4). Furthermore, there was 36.19 and 39.35 s of after-glow time. However, all the FRT samples were self-extinguished after removing the combustion source (Table 2). Moreover, the graph of Figure 3 also shows that the after-flame time of FRT samples were shorter than two s except for the sample treated at 160 °C for 60 s. In addition, there were char-forming on the sample areas exposed to the flame (Figure 4). Moreover, the char-length of the FRT samples also varied depending on the curing conditions, but the difference was not of importance. The results show that the higher the curing temperature, the longer the curing time is, the shorter the char length of the FRT samples is (Table 2). Although there is no fitted model between char-length and 2 factors A and B, their values have also varied relatively according to the changes of these factors.

Therefore, the shortest char length corresponds to the sample treated at 180 °C for 120 s, which completely corresponds to the highest LOI value.

#### 3.3.3. Mechanical Strength Loss of FRT Cotton Fabric

The mechanical strength losses of the FRT samples in comparison with the control fabric are presented in Table 6.

In general, these losses are quite important, the smallest is 32.44% and the highest is 47.32%. The effects of the temperature and time of the curing step on the mechanical strength loss in some cases are quite obvious. When the samples were cured at 160 and 180 °C, the longer the curing time is, the higher this loss is. Similarly, when the curing time was for 90 and 120 s, the higher the curing temperature is, the higher this loss is. In the other conditions, the above trend is not yet clear. The loss of the mechanical strength of cotton fabric after flame retardant treatment with PCN has been recorded in many works [2,6,7,8,9,10,11,12,13]. However, in this study, this loss up to 47% can be considered as too high in comparison with the results of our previous study [14], in which, the flame-retardants treatment for cotton fabrics were carried out using pad–dry-cur method without plasma activation, the mechanical strength loss was only from 17.7 to 32.4%. Thus, the higher mechanical strength loss of the FRT samples in this study could be only attributed to the plasma activation step. In addition, the mechanical strength loss of cotton fabric due to plasma pretreatment has also been noted in some works [19,21]. This hypothesis will be proved in the following section by the morphologic analysis of the cotton fiber surface.

#### 3.3.4. Optimizing the Temperature and Time of the Curing Step

As the aim of this study was to get the fabric having the LOI ≥ 25 and the mechanical strength loss of the treated fabric due to the curing step is as low as possible. According to these criteria, Design-Expert software has given the optimal temperature and time of curing step and predicted the LOI value, the F_max_ in the warp direction of the cotton fabric if it would be treated in this condition (Table 7)

Table 7 shows that if the goal was just LOI > 25, the optimal curing condition would be at 160 °C for 90 s with the desirability index of 0.920. If APDBD plasma-treated cotton fabric was treated under this condition, it would have the LOI of 25.0 and the warp F_max_ of 979 N equivalent the mechanical strength loss of 35%. The actual tests at predicted optimal conditions have shown that the LOI value of the treated fabric was 26.8 and its warp F_max_ is reduced by 34.9% in comparison with the control fabric. Compared to the traditional pad–dry-cure method [14], with the same purpose LOI > 25, the optimal curing condition was predicted at 180 °C for 114 s, the predicted LOI value would be 25.5. However, the mechanical strength loss of the FRT fabric was predicted to be only 30.29%. Thus, in the flame-retardant treatment process with plasma-activation, although the curing step was carried out at lower temperatures for a shorter duration, the mechanical strength loss would be higher. The following fiber surface analysis will allow a better understanding of this phenomenon.

### 3.4. Results of the Fiber Surface Analysis.

#### 3.4.1. Scanning Electron Microscopy Images

The SEM images of the fiber surfaces of the control sample, plasma-treated sample and FRT samples are shown in Figure 5 and Appendix A. On the surface of the control fiber (Figure 5c), the fibril layer on a spiral pattern was observed, but the surface is quite smooth. The surface morphology of the plasma-treated fibers (Figure 5a,b) are completely different from the untreated fiber. The rugged fiber surface with many spiral grooves adjacent to each other along the fiber length. In addition, there are dense spots on the groove surface. Furthermore, a comparison between Figure 5a,b shows that the grooves on the fiber surface tend to be more continuous and wider for the sample with the plasma exposure time of 90 s. Figure 5d–f and Appendix A display a uniform surface coating. The grooves on the fibrous surface are almost no longer visible. From the SEM micrographs assessment Figure 5d–f and Appendix A, it is clear that the FR formulations (PCN and K-FFRC) were deposited evenly on the fiber surface. In order to further support this finding, phosphorus (P) and nitrogen (N) mapping was carried out using EDS.

#### 3.4.2. Energy-Dispersive Spectroscopy

The EDS spectra, N and P mapping of the control sample, APDBD sample and FRT samples are shown in Figure 6 and Appendix A. The first column on the left shows the EDS N mapping, followed by the EDS P mapping. The EDS spectra of the samples are presented in the far right. The atomic content of the various elements in the samples were determined from the EDS spectra and are shown in Table 8.

As expected, the control and plasma-treated samples had a large presence of carbon and oxygen atoms and there are no peaks N, P on their EDS spectra (Figure 6a,b, Table 8). However, in the FRT samples, the atomic peaks corresponding to nitrogen and phosphorus are easily visible in the EDS spectra of Figure 6c–e and Appendix A. In addition, it can be noticeably seen from P and N mapping in Figure 6c–e and Appendix A that the coating covers the fiber surface uniformly, and the P and N distribution was homogenous.

Table 8 shows that all the FRT samples has quite high phosphorus content. The lowest content is 1.29% for the sample treated at 160 °C for 60 s. The highest phosphorus content reaches 1.52% for the sample treated at 180 °C for 120 s. Moreover, the phosphorus content of the samples is also related to the vertical flammable characteristics and the LOI values of the samples. The sample treated at 160 °C for 60 s has the highest after-flame time, lowest LOI value and lowest phosphorus content. All the samples with the LOI higher than 25.0 have the phosphorus content near and higher than 1.4%. Other studies have also used EDS spectra, EDS N mapping, EDS P mapping to prove the presence of PCN on the fabric treated with PCN [8,11].

The EDS spectra and N and P mapping confirm that the coating on the fiber surface observed in Figure 6c–e and Appendix A is PCN and K-FFRC.

#### 3.4.3. X-ray Photoelectron Spectrometer

The chemical composition of the control and plasma-treated samples for the different exposure time investigated by XPS are presented in Figure 7. Table 9 shows the functional groups in the form of atomic concentration. According to Figure 7 and Table 9, the untreated cotton (Figure 7a) exhibits three functional groups, which are the cellulosic structure (C–C/C–H, C–OH/C–O and O–C–O/C=O groups). For the plasma-treated samples (exposure time for 45 s in Figure 7b and for 90 s in Figure 7c), similar to the untreated sample, three functional groups (C–C/C–H, C–O/C–OH and O–C–O/C=O groups) are also observed on the surface. However, in comparison with the untreated fabric, the amount of the C–C/C–H bonds on the treated surface decreased significantly (less than 17.3% for the exposure time of 45 s and 15.1% for the exposure time of 90 s). The amount of C–O/C–OH has almost unchanged for the exposure time of 45 s but increased 7% for the exposure time of 90 s. The amount of O–C–O/C=O increased almost twice for the exposure time of 45 s and increased by 6.18% for the exposure time of 90 s. Moreover, the plasma treatment for 90 s introduced a new O=C–O functional group on the surface. In general, under the influence of atmospheric plasma, C–C/CH groups decreased while the oxygen groups increased and especially the new group O=C–O appeared. This phenomenon was also observed in [30,32,33,34] when the cotton fabric was treated with oxygen plasma. Thus, it is evident from XPS analysis that the APDBD plasma treatment has increased the oxygen functional groups, it made the cotton fabric more active, promoting the reaction between cellulose and FRs.

## 4. Discussion

By comparing the SEM images (Figure 5a–c and the XPS spectra (Figure 7, Table 9) of the cotton samples before and after plasma treatment, it can be said that there were the chemical and physical modifications of the material surface. Maybe, the interaction of excited species (in the plasma environment) with the solid surfaces placed in the plasma (cotton fibers) is the cause of these modifications. The active species react with the substrate surface, this creates chemical functionality on the substrate surface. Moreover, the ablation of materials by plasma can occur by two principal processes, one is physical sputtering and the other is chemical etching. The sputtering of materials by chemically non-reactive plasma, this is a knock-on process by ions with high energy [18]. Chemical etching occurs in chemically reactive types of plasma, during etching reaction, weight loss of the substrate occurs and the topmost layer of the substrate is stripped off (Figure 5a,b). The weight loss in the etching process is mainly due to bond scission of polymers and to reactions of the radicals generated in the polymer chains of the substrate by the plasma exposure [18].

Thus, there are both physical and chemical modifications on the plasma-treated cotton fiber surface, resulting in more active cotton fibers. Therefore, the plasma-treated cotton samples were more easily chemically bound to the PCN and K-FFRC. As a result, in the process of flame retardants treatment, plasma-treated cotton fabrics only needed to be cured at 160 °C for 90 s to have the same flame-retardant effect as those of un-treated cotton fabrics when it was cured at 180 °C for 114 s [14].

However, plasma treatment also caused the weight loss and the bond scission of polymers, as mentioned above, especially the addition of a new functional group COOH (Figure 7, Table 9). These modifications explain why the tensile breaking strength of the cotton fabric was reduced by 27% after 90 s of plasma treatment. They also explained why the curing condition in the flame-retardant treatment of plasma-treated cotton samples was only 160 °C for 90 s, but the loss in mechanical strength of FRT fabrics was 35% (Table 7), this is the mechanical strength loss of both processes (plasma and curing steps). While, the curing condition for the non-plasma-treated cotton fabric was 180 °C for 114 s, but the loss in tensile strength of flame-retardant fabric is only 30.29% [14], this is only the mechanical strength loss of the curing step. Therefore, to reduce this loss, the exposure time of plasma pretreatment seems to need to be shorter. It needs to be so short that during this time, the plasma pretreatment can only play a supporting role in creating chemical bonds between cellulose and chemicals without any mechanical damage to the fabric. This issue will be studied in our next work.

## 5. Conclusions

From the research results, the following conclusions can be drawn:(1)Applying APDBD plasma with the power of one watt per square centimeter to cotton fabric for 90 s before it is treated with FR can allow a significant reduction in temperature and time of curing step. However, the FRT fabric is always flame retardant fabric. This could allow to minimize the mechanical strength losses to cotton fabrics caused by the curing step. Nevertheless, the plasma power of one watt per square centimeter and the exposure time of 90 s also caused the considerable mechanical damage to the cotton fabric. In this case, the mechanical damage to the fabric of both causes may be greater than in the absence of plasma pretreatment. Therefore, it is necessary to find the optimal plasma parameters so that it can support the bond between the chemicals and the fabric, but its damage to the mechanical strength of the fabric is as low as possible. Thus, plasma pretreatment exposure time should be so short that the plasma pretreatment can only support the creation of chemical bonds between cellulose and chemicals without damaging their mechanical strength.(2)How to reduce the plasma exposure time and no need to increase the temperature and time of the curing step, but the FRT fabrics always have the LOI ≥ 25? The content of this study will be detailed in our next work.

## Figures and Tables

**Figure 1 polymers-12-01575-f001:**
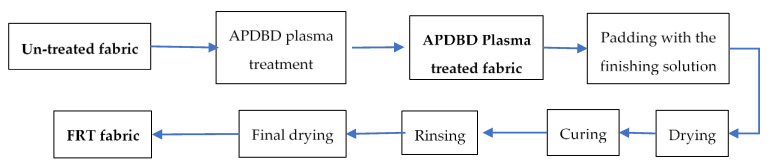
Treatment procedure for cotton fabric.

**Figure 2 polymers-12-01575-f002:**
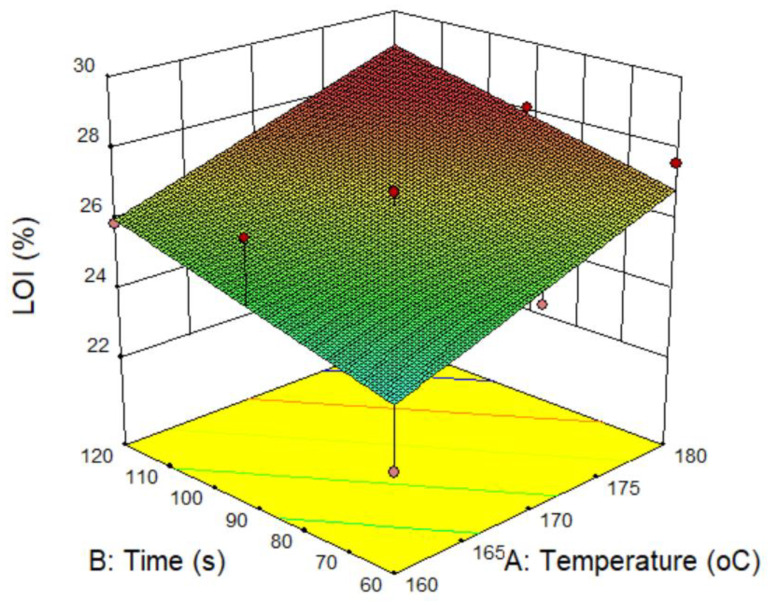
Surface response curve related to limiting oxygen index (LOI) of samples.

**Figure 3 polymers-12-01575-f003:**
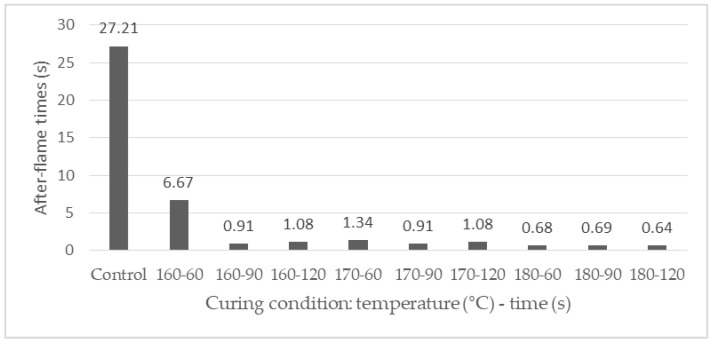
After-flame time of the control and flame retardant-treated (FRT) samples under ASTM D6413 test.

**Figure 4 polymers-12-01575-f004:**
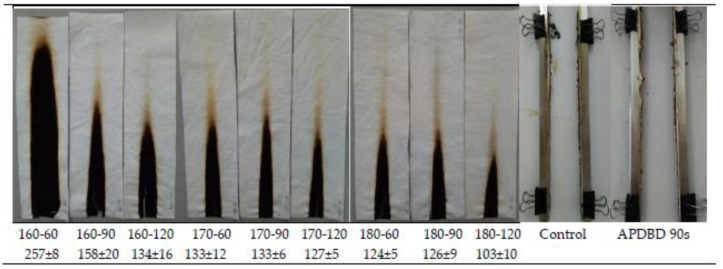
Images of the FRT, control and atmospheric-pressure dielectric barrier discharge (APDBD) plasma-treated samples after the vertical flammability test. Upper line: curing condition: temperature (°C)—time (s); bottom line: char length (mm).

**Figure 5 polymers-12-01575-f005:**
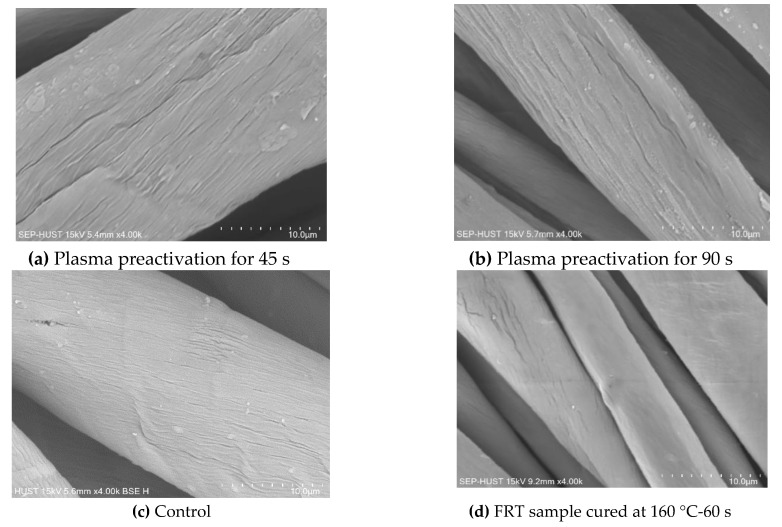
SEM images of the APDBD plasma-treated samples, control and some FRT samples at a magnification of 4000×. (**a**,**b**) APDBD plasma-treated samples with the exposure time of 45 and 90 s, respectively; (**c**) control sample; (**d**–**f**) samples cured at the different curing condition (curing temperature (°C)– curing time (s)).

**Figure 6 polymers-12-01575-f006:**
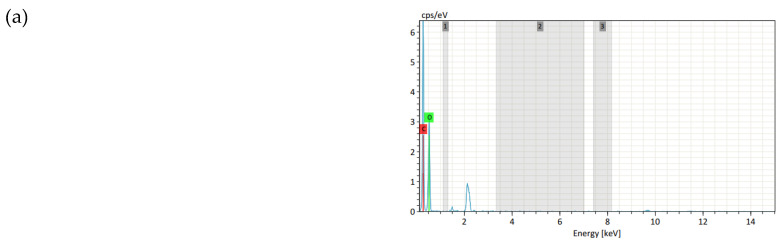
EDS N, P mapping and elemental spectral profile of (**a**) Control sample; (**b**) plasma-treated sample for 90 s; (**c**) FRT sample cured at 160 °C for 60 s; (**d**) cured at 160 °C for 90 s; (**e**) cured at 180 °C for 120 s.

**Figure 7 polymers-12-01575-f007:**
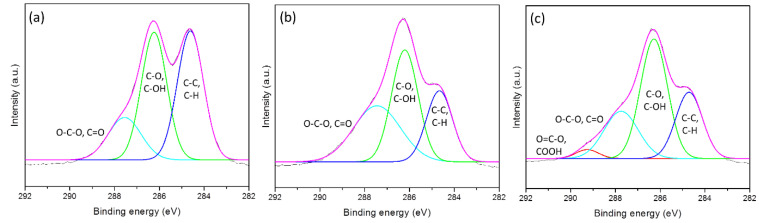
Deconvolution of C1s spectra for (**a**) control; (**b**) plasma-treated for 45 s; (**c**) plasma-treated for 90 s.

**Table 1 polymers-12-01575-t001:** Central composite designs type face centered (CCF) design of experiments.

Exp. N^o^	A	B	X_1_ (°C)	X_2_ (s)
1	+1	−1	180	60
2	+1	+1	180	120
3	0	−1	170	60
4	−1	0	160	90
5	0	0	170	90
6	+1	0	180	90
7	0	0	170	90
8	0	+1	170	120
9	−1	+1	160	120
10	−1	−1	160	60

**Table 2 polymers-12-01575-t002:** CCF experimental design and results.

Sample	Variable Factor		LOI (%)	Characteristics of Vertical Flammability Test			Warp Fmax (N)	Mechanical Strength Loss (%)
	Curing Temp. (°C)	Curing Time (s)		After-Flame Times (s)	After-Glow Times (s)	Char Length (mm)		
Control (CS)	–	–	17.1	27.21	36.19	Completely burned	1507 ± 43	
Plasma-treated (PS)	–	–		28.03	39.35	Completely burned	1097 ± 71	27.18
1	180	60	27.6	0.68	0	124 ± 5	1018 ± 39	32.44
2	180	120	28.0	0.64	0	103 ± 10	794 ± 47	47.32
3	170	60	25.0	1.34	0	133 ± 12	936 ± 77	37.90
4	160	90	26.8	0.91	0	158 ± 20	981 ± 52	34.91
5	170	90	26.8	0.91	0	133 ± 6	952 ± 69	36.82
6	180	90	28.1	0.69	0	126 ± 9	890 ± 66	40.96
7	170	90	26.8	–	–	–	854 ± 112	43.35
8	170	120	27.2	1.08	0	127 ± 5	969 ± 13	35.71
9	160	120	25.9	1.08	0	134 ± 16	976 ± 39	35.22
10	160	60	22.1	6.67	0	257 ± 8	1003 ± 83	33.42

**Table 3 polymers-12-01575-t003:** Model fitting of test result.

Test Response	Model Parameter				Response Equation in Actual and Code Variables
	R- Squared	Adj R- Squared	F Value	*P**-* Value	
LOI	0.6978	0.6115	8.08	0.0152	$Y1=-1.987+0.148X_{1}+0.036X_{2}$ Z1=26.43+1.48A+1.07B

**Table 4 polymers-12-01575-t004:** Shorted ANOVA of the model.

Source	F-Value	*p*-Value Probe > F
***Model***	***Y1***	
$X_{1}$-Temperature	10.66	0.0138
$X_{2}$-Time	5.51	0.0513

**Table 5 polymers-12-01575-t005:** Estimated conditions of the curing step allowed LOI ≥ 25.

Option	Temperature (°C)	Time (s)
1	160	90
2	167	63

**Table 6 polymers-12-01575-t006:** Mechanical Strength loss of the FRT sample according to the temperature and time of the curing step.

Loss in Tensile Strength (%)			
Curing Time (s)	Curing Temperature (°C)		
	160	170	180
**60**	33.42	37.90	32.44
**90**	34.91	36.82	40.96
**120**	35.22	35.71	47.32

**Table 7 polymers-12-01575-t007:** Optimizations of curing conditions.

Number	Temperature (°C)	Time (s)	LOI	Warp F_max_ (N)	Desirability	Goal
1	160	90	25.0	979	0.920	Temperature minimize, time 60->90 s, LOI > 25, Tensile strength maximize

**Table 8 polymers-12-01575-t008:** The atomic percentage of different elements present in the samples were determined from the EDS spectra.

Samples	C (%)	O (%)	P (%)	N (%)
(a) Control	56.62	43.38	-	-
(b) Plasma-treated for 90 s	56.51	43.49	-	-
(c) Cured at 160 °C-60 s	54.25	40.56	1.29	3.91
(d) Cured at 160 °C-90 s	55.60	39.93	1.49	2.98
(e) Cured at 160 °C-120 s	53.64	40.50	1.36	4.51
(f) Cured at 170 °C-60 s	54.94	40.01	1.53	3.52
(g) Cured at 170 °C-90 s	55.31	40.48	1.51	2.70
(h) Cured at 170 °C-120 s	54.09	41.09	1.36	3.46
(i) Cured at 180 °C-60 s	53.92	39.85	1.42	4.84
(j) Cured at 180 °C-90 s	54.58	40.33	1.48	3.61
(k) Cured at 180 °C-120 s	54.06	40.30	1.52	4.12

**Table 9 polymers-12-01575-t009:** Percentage of the chemical groups present on the surface of control and plasma-treated cotton.

Plasma Exposure Time (s)	Functional Group (%)			
	C–C/C–H	C–O/C–OH	O–C–O/C=O	O=C–O/COOH
0	42.10	40.40	17.50	–
45	24.84	39.80	35.35	–
90	26.05	47.34	23.68	2.93

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
