# Peer review of "Application of Plasma Activation in Flame-Retardant Treatment for Cotton Fabric"

_polymers, 2020, doi:10.3390/polym12071575_

Round 1

Reviewer 1 Report

Manuscript entitled “Application of plasma activation in flame-retardant treatment for cotton fabric” involves a flame-retardant cotton fabric, applying the atmospheric-pressure DBD plasma on the cotton fabric and treating flame retardants on it. The overall quality of this manuscript suits to Polymers, however, in order to be considered as publication, major revision will be needed for further consideration.

1 The abstract is prolix. The authors should summarize it in brief.

2 Introduction is well written. However, add more literature related with fire retardant in fireproof issues. Cite these relevant references in introduction: (1) X. Hu, X. Zhu, Z. Sun, Effect of CaAlCO3-LDHs on fire resistant properties of intumescent fireproof coatings for steel structure, Applied Surface Science 457 (2018) 164-169. (2) X. Hu, X. Zhu, Z. Sun, Efficient flame-retardant and smoke-suppression properties of MgAlCO3-LDHs on the intumescent fire retardant coating for steel structures, Progress in Organic Coatings 135 (2019) 291-298. (3) X. Hu, Z. Sun, X. Zhu, Z. Sun, Montmorillonite synergized water-based intumescent flame retardant coating for wood, Coatings 10 92020) 109.

3 The experimental setup of Atmospheric Pressure DBD Plasma Treatment for Fabric should be supplemented in Materials and Methods.

4 The abbreviations in full text are confusing. They are should be abbreviated under the same rule.

5 Results and Discussion should be put together, rather than devided.

6 Conclusion is missing in the paper.

Reviewer 2 Report

This paper concerns the flammability of cotton fabric using Plasma treatment. The characterization of the system is adequate. I think that this paper deserves to be published but some improvements are required.

  1. The manuscript should be carefully checked with language-proof. There are still some typos and grammar issues to be corrected.
  2. The abstract is too long. The full names of abbreviations should be given, such as FFRC, LOI.
  3. In introduction part, the authors just explain why this work did, and the novelty of this work should also be provided.
  4. The authors should provide the variability in the Warp tensile strength of each samples.
  5. As the authors pointed out, the purpose of this work is to have a flame-retardant cotton fabric (LOI≥ 25) and the mechanical loss of the treated fabric due to the curing step is as low as possible. The temperature and curing time reduced, but the tensile tensile strength decreased seriously. And the authors think it is still necessary to find the optimal plasma parameters, so what is your plan? I think this part is very important, and should be provide in detail in the Discussion part.

Reviewer 3 Report

Dear Authors,

congratulation to this paper.

I have four remarks:

1) A LOI value has five digits. This is to much. I know it is the calculated value. Please reduce the digits to 3.

2) first line page 2: you had ".," Pleas correct this.

3) Figure 6 has a lot of pictures with the same information. Perhaps you show one or two examples and add the other ones to the supplement. This remark is for Figure 6 first and second. You have Figure 6 twice in the paper.

4) Table 8 is before table 7.

BR

The reviewer

Round 2

Reviewer 1 Report

The authors have answered all questions in detail. The paper can be published in Polymers.

Reviewer 2 Report

The authors have solved all the questions.